A new volute, Ericusa ngayawang sp. nov. (Gastropoda: Volutidae), from the Miocene of South Australia

Yates Adam M. adamm.yates@magnt.net.au
Museum and Art Gallery of the Northern Territory , Alice Springs, Northern Territory , Australia
Rosenberg Gary
Electronic publication date: 2022 Oct 5
Publication date: 2022
Volume: 10
Electronic Location ID: e14197
Received 2022 Feb 25; Accepted 2022 Sep 15
Copyright: © 2022 Yates
Copyright year: 2022
Copyright holder: Yates
License: This is an open access article distributed under the terms of the Creative Commons Attribution License, which permits unrestricted use, distribution, reproduction and adaptation in any medium and for any purpose provided that it is properly attributed. For attribution, the original author(s), title, publication source (PeerJ) and either DOI or URL of the article must be cited.
License URL: https://creativecommons.org/licenses/by/4.0/

Keywords: Volutidae, Murray basin, Miocene, Cadell formation, South Australia, Southeast Australian Province, Langhian, Biogeography, Gastropoda

Funding: The author received no funding for this work.

==============================
Ericusa ngayawang sp. nov. is described from shells preserved in the Middle Miocene Cadell Formation in the western Murray Basin of South Australia. At the time the Murray Basin was part of the Southeastern Australian Marine Biogeographic Province. Ericusa ngayawang is a small heavily costate species of Ericusa with clear affinities to the Early Miocene E. atkinsoni of Victoria and Tasmania but can be distinguished from it by its smaller size, more slender proportions and its heavily costate body whorl. Ericusa atkinsoni and its relative, E. macroptera, inhabited the basins to the east of the Murray Basin during the Late Oligocene and Early Miocene but were extinct there before the end of the Burdigalian Stage of the Early Miocene. The persistence of E. ngayawang into the Langhian Stage of the Middle Miocene is another piece of evidence for partial biogeographic isolation of the western Murray Basin from the rest of the Southeastern Australian Province during the Miocene.

Introduction

Ericusa Adams & Adams (1858), is an endemic Australian genus of volutid that is characterized by a fusiform shell and a moderately large to large, paucispiral protoconch that has its axis of coiling offset from that of the teleoconch by about 45° or less (Darragh, 1989). The oldest named species of the genus come from the Chattian Stage, Late Oligocene of the Jan Juc Formation in the Otway Basin of Victoria. Two distinct species are present in this formation, indicating that the genus has an older origin and that some diversification had occurred prior to the deposition of the Jan Juc Formation. Indeed, Darragh (1989) reported fragments of an undescribed volutid from the Priabonian Stage (Late Eocene), Narrawaturk Marl (as Browns Creek Clay) that bear similarity to Ericusa atkinsoni. These fragments have not been figured so it is not possible to assess whether or not they truly belong to Ericusa. There is a distinct possibility that they may belong to a stem-member of the tribe Livoniini to which Ericusa belongs (Bail & Poppe, 2001). Among the early occurring species of Ericusa, E. atkinsoni (Pritchard, 1896) and its relative E. macroptera (McCoy, 1866) stand out from extant species in possessing a strongly exsert tip of the first protoconch whorl and axial sculpture in form of costae, often with shoulder nodules, on at least the early teoloconch whorls. These two species bear some resemblance to early members of Livonia, particularly L. stephensi (Johnston, 1880). Livonia is the presumed sister genus of Ericusa (Darragh, 1989) and the resemblance between L. stephensi and especially E. atkinsoni is probably an indication that neither is particularly distant from the common ancestor of the two genera. Thus it is plausible that E. macroptera and E. atkinsoni are an early branch of the genus and may well be the sister group to all other species in the genus. The E. macroptera–E. atkinsoni lineage was short lived in south eastern Australia and makes its last appearance before the end of the Burdigalian Stage (Early Miocene) in the Fishing Point Marl of the Otway Basin (Darragh, 1989).

Here a new species of the E. macroptera–E. atkinsoni lineage is described from the Langhian Stage (middle Miocene) of the more westerly Murray Basin of South Australia. It indicates that the lineage survived into the Middle Miocene after going extinct in the more easterly Otway and Bass Basins of Victoria and Tasmania. It is further evidence of the biogeographic distinctiveness of the molluscan fauna of the Murray Basin in comparison to more easterly basins during the Miocene.

Geological setting

All of the specimens of the new species described here were collected from the Murbko Marl Member of the Cadell Formation of the Morgan Subgroup (Lukasik & James, 1998; Cowley & Barnett, 2007), at its type section on the east bank of the Murray River, 6 km south of Morgan, South Australia (Fig. 1). The Morgan Subgroup is a part of the Oligo-Miocene Murray Group, a marine sequence filling the western Murray Basin (Ludbrook, 1961; Lukasik & James, 1998; Gallagher & Gourley, 2007). The Murbko Marl Member is a soft grey marl containing an abundant and diverse mollusc assemblage (Lukasik & James, 1998). Much of the Murray Group is composed of porous calcarenites where ground water has stripped away original aragonite, leaving only moulds of aragonitic shells. The Murbko Marl Member is one of very few beds where this has not occurred, making it a favourite destination for amateur collectors and professional palaeontologists alike.

Figure 1 Locality map of the occurences of Ericusa ngayawang sp. nov. and their position within the western Murray Basin of South Australia.

Fossil localities with E. ngayawang are marked with stars, the type locality is arrowed. Inset shows the position of the main map (marked as a box) within the Murray Basin (marked as solid light brown colour).

Outcrops of the Cadell Formation are restricted to the western part of the Murray Basin, in the vicinity of the town of Morgan. Nevertheless subsurface argillaceous beds that can be correlated with the Cadell Formation extend eastwards into far western Victoria (Gallagher & Gourley, 2007).

The age of the Cadell Formation is early middle Miocene (Langhian Stage) based on biostratigraphy using planktonic foraminifera (Li & McGowran, 1999).

Materials and Methods

The specimens described herein were all collected from the Cadell Formation at its type section. Collection of fossil shells in South Australia is considered “fossicking” and can be conducted on non-reserved public lands without a permit provided no mechanical devices or explosives are used and the specimens are not sold. Most specimens were found in the loose slips of eroded material that cover much of the slope of the Cadell Formation at its type locality but the holotype was excavated from the cliff section, at approximately two thirds of the height of the formation from its base. These specimens have been deposited in the palaeontological collection of the Museum and Art Gallery of the Northern Territory, held at Megafauna Central, in Alice Springs, Northern Territory.

The electronic version of this article in Portable Document Format (PDF) will represent a published work according to the International Commission on Zoological Nomenclature (ICZN), and hence the new names contained in the electronic version are effectively published under that Code from the electronic edition alone. This published work and the nomenclatural acts it contains have been registered in ZooBank, the online registration system for the ICZN. The ZooBank LSIDs (Life Science Identifiers) can be resolved and the associated information viewed through any standard web browser by appending the LSID to the prefix http://zoobank.org/. The LSID for this publication is: urn:lsid:zoobank.org:pub:0CFA9C77-D089-49D5-A6FA-EA51099D57E6. The online version of this work is archived and available from the following digital repositories: PeerJ, PubMed Central SCIE and CLOCKSS.

Systematic palaeontology

GASTROPODA Cuvier (1795)

CAENOGASTROPODA Cox (1960)

NEOGASTROPODA Wenz (1938)

VOLUTIDAE Rafinesque (1815)

CYMBIINAE Adams & Adams (1858)

LIVONIINI Bail & Poppe (2001)

ERICUSA Adams & Adams (1858)

Type species—Voluta fulgetrum Sowerby (1825)

ERICUSA NGAYAWANG SP. NOV.

urn:lsid:zoobank.org:act:E133FA7E-2327-4AA3-8D55-B7B9DA5CF950

Etymology—Named after the now extinct language spoken in the area that the fossils were found prior to European colonisation (Horgen, 2004). There are few words of this language recorded, so it seems appropriate that the people that spoke this language are at least commemorated here.

Holotype—NTM P11217, an adult shell missing the anterior end of the body whorl (Fig. 2).

Figure 2 Type specimens of Ericusa ngayawang sp. nov.

(A–C) Holotype (NTM P11217) in (A) ventral, (B) labial and (C) dorsal views. (D, E) Juvenile paratype (NTM P8464) in (D) dorsal and (E) ventral views. (F) Close-up view of the protoconch of the holotype (NTM P11217). Scale bar for A–E = 20 mm. Scale bar for F = 5 mm. Specimens have been whitened with ammonium chloride. Photographs by the author.

Paratypes—NTM P8464, juvenile; NTM P9019, three spire fragments.

Type locality and horizon—Cliffs immediately south of small gully, 4.8 km South of Morgan Ferry–Cadell Road on the east bank of the Murray River, opposite Brenda Park (The type section of the Murbko Marl Member of the Cadell Formation). Langhian Stage, Miocene.

Specific diagnosis—Adult size approximately 70 mm. Shell slender with gradually tapering spire. First whorl of protoconch with exsert tip and deviated 40° to 15° from axis. Axial costae persistent from first teleoconch whorl to body whorl. Reflexed outer lip narrow, not winglike.

Description—The shell is small for an Ericusa (Fig. 3; Table 1). With an estimated adult length of approximately 70 mm, the holotype lies within the known size range of Ericusa subtilis, and exceeds only E. naniforma (Bail & Limpus, 2013). The protoconch consists of 2 to 2.5 smooth whorls. The first of these is rounded and deviated from the axis of coiling of the rest of the shell. The angle of deviation is less than that observed in other species of Ericusa and varies from 40° to 15°. The initial portion of the first protoconch whorl forms a low rounded posterior projection that resembles a more subdued version of the more strongly exsert tip seen in E. atkinsoni and E. macroptera. The adult spire consists of about four whorls with convex profiles. Each spire whorl bears 11 to 12 rounded axial costae that become weakly nodulose on the last half of the last whorl. The spire whorls are crossed with fine spiral threads that attenuate on the last spire whorl although a few weak threads below the shoulders do persist onto the early part of the adult body whorl. There is a narrow and weakly concave posterior whorl slope between the shoulder nodules and the posterior suture. The middle section of the body whorl is mildly ventricose and expands slightly lateral to the level of the shoulder nodules. The anterior end is abruptly contracted and lacks a siphonal fasciole. The axial costae continue to be expressed on the body whorl with only the last example before the outer lip being weakly developed. The costae are weak to obsolete on the posterior whorl slope and bear moderately developed shoulder nodules. Each costa extends anteriorly to about the level of the anterior contraction, whereafter they become obsolete. The adult outer lip is weakly laterally everted and bears a small posterior expansion that is deflected dorsally. The columella is covered with a thin callus and bears three strong columellar plaits. No residual colour pattern could be observed under UV light.

Figure 3 Shells of various Livoniini compared.

(A) Silhouettes from left to right of Livonia stephensi, Ericusa macroptera, Ericusa atkinsoni, Ericusa ngayawang sp. nov. and Ericusa fulgetrum drawn to scale. (B–F) Reconstructions of adult shells from each species showing diagnostic features (not to scale). (B) Livonia stephensi. (C) Ericusa macroptera. (D) Ericusa atkinsoni. (E) Ericusa ngayawang sp. nov. (F) Ericusa fulgetrum.

Table 1 Dimensions of Ericusa ngayawang sp. nov. and other selected species of Ericusa.

Length measured from the protoconch to the end of the anterior canal along the axis of the shell. Spire height measured from the level of the protoconch to the level of the apertural suture, parallel with the axis of the shell. For those measurements taken from the literature spire height is equal to length minus aperture height. Width measured perpendicular to the axis of the shell at its widest point, between axial costae. Where two measurements appear the specimen is incomplete and the number in parentheses refers to the measurement as preserved, whereas the number not in parentheses is the estimated measurement if complete.

	Length (mm)	Spire height (mm)	Width (mm)	
Ericusa macroptera NMV P12379*	125	–	62	
Ericusa macroptera NMV P12378*	141	36	64	
Ericusa macroptera NMV P48588*	134	47	52	
Ericusa atkinsoni NMV P9985*	132	47	66	
Ericusa atkinsoni NMV P41723*	140	48	61	
Ericusa ngayawang NTM P11217	~70 (59)	30	27	
Ericusa ngayawang NTM P8464	30	9	16	
Ericusa subtilis WAM 69.515*	67	29	24	
Ericusa subtilis WAM 79.391*	71	30	25	
Ericusa naniforma WAM S11656$	62	18	34	
Notes:

* Measurements taken from Darragh (1989).

$ Measurements taken from Bail & Limpus (2013).

Specific comparisons—Differs from all species of Ericusa except E. atkinsoni and E. macroptera in its possession of both axial sculpture and an exsert tip of the protoconch. It differs from E. macroptera in the absence of a large, wing-like posterolateral expansion of the outer lip and its far stronger axial sculpture that persists onto the body whorl. It differs from E. atkinsoni by its smaller size (adult length approximately 70 mm vs. 132–140 mm: Darragh, 1989), the persistence of elongate axial costae onto the body whorl (vs. restriction of axial sculpture to short shoulder nodules on the body whorl), and its overall more slender shell with a less tumid body whorl and a relatively longer, more gradually tapering spire. It differs from Livonia stephensi in its smaller size, more slender form and absence of an exsert tip on the first protoconch whorl (Fig. 3).

Discussion

Ericusa ngayawang is clearly related to E. atkinsoni. The latter species is known from the Freestone Cove Sandstone of the Bass Basin, Tasmania and from the Puebla Formation of the Torquay Sub-Basin of the Otway Basin in Victoria (Darragh, 1989; Fig. 4). These two formations considered to be age equivalent (Ludbrook, 1967) and both contain Darragh (1985) molluscan assemblage VIII. Darragh’s numbered molluscan assemblages are a chronological series of assemblage zones that divide the Cenozoic marine sequence of southeastern Australia. The age of the Puebla Formation is well-constrained to the earliest Miocene (Aquitanian Stage) both with microfossil biostratigraphy (Li, Davies & McGowran, 1999) and strontium isotope stratigraphy (Dickinson, 2002 reported in McLaren et al., 2009).

Figure 4 Palaeobiogeography and stratigraphy of selected early Livoniini relevant to this study.

Timescale from Cohen et al. (2013, updated 2020). Stratigraphic columns show known time ranges of species within each relevant basin. Maps show geographic occurrences of species (colour coded to stratigraphic columns) in south-eastern Australia for three stages (Chattian, Aquitanian and Langhian). Dark blue in all maps represents present day seas, tan represents present day land and light blue indicates maximum extent of Cenozoic marine transgression over present day land.

Younger specimens of E. atkinsoni are also known from the Fishing Point Marl (Darragh, 1989; Fig. 4). This unit, which occurs in the main Otway Basin, contains Darragh (1985) molluscan assemblage IX, and includes a number of first occurrences of younger taxa not found in the Freestone Cove Sandstone, Puebla Formation, or older beds, indicating that it is younger than these. Carter (1958) placed the lower part of the Fishing Point Marl in his foraminifera zone G which correlates with the mid part of the Burdigalian Stage of the Early Miocene (Gallagher & Stanislaus, 2019).

Darragh (1989) did not figure the Fishing Point Marl specimens but did mention that they are both smaller and more slender with elongate spires than topotype specimens. In these respects they show similarity to E. ngayawang and it is possible that the Fishing Point Marl specimens represent an intermediate population or even an early population of E. ngayawang. However Darragh (1985) indicated that the Fishing Point Marl specimens were few in number and of poor preservation, hindering a definitive systematic assessment.

Following the deposition of the Fishing Point Marl in the mid Burdigalian Stage, the Miocene climatic optimum (Böhme, 2003) was reaching its zenith and transgressions covered the onshore basins of south eastern Australia for much of the remaining Early and Middle Miocene (McGowran et al., 2004). As a consequence there are many marine mollusc fossil sites dating to the late Early Miocene and Middle Miocene that were traditionally assigned the local stage names Batesfordian, Balcombian and Bairnsdalian, including the famously rich sites of Muddy Creek and Fossil Beach, Balcombe Bay. None of these have yielded an Ericusa that can be referred to the E. macroptera–E. atkinsoni lineage and it is reasonable to infer that the lineage was extinct east of the Murray Basin before the end of the Burdigalian Stage.

Mention should be made of a specimen retrieved from a depth of 73 m in Mundys Well in the Murray Basin on Canegrass Station 70 km NNE of the type section for the Murbko Marl Member of the Cadell Formation (Fig. 1). This specimen was referred to E. atkinsoni by Darragh (1989) but an examination of the specimen, well over a decade ago, by myself indicated that it was the same species as the one present in the Cadell Formation. Unfortunately the specimen could not be re-located in the collections at the South Australian Museum for measurement and figuring in this article. The stratigraphic position of the fossils from Mundys Well has never been discussed. Other molluscs from the same level in the well include Corbula ephamilla (Tate, 1885) and the nominate subspecies of Athleta (Ternivoluta) antiscalaris (McCoy, 1866) which are common members of the Cadell Formation assemblage at the type section for the Murbko Marl Member. Gallagher & Gourley (2007) found that the Cadell Formation was extensive in subsurface sections and could be recognised in several boreholes in western Victoria. Given the proximity of Mundys Well to the type section of the Murbko Marl Member of the Cadell Formation, the presence of other mollusc species that are typical of the Cadell Formation and the broad subsurface extent of the Cadell Formation, there is little doubt that the mollusc fossils from Mundys Well were taken from the Cadell Formation, nor is there much doubt that the specimen in question can be referred to E. ngayawang.

The majority of mollusc species in the Cadell Formation are shared with middle Miocene mollusk assemblages from the Otway and Port Phillip Basins. In particular many species are shared with Darragh (1985) molluscan assemblage XI which is typified by the famously rich deposit at Fossil Beach, Mornington Peninsula in the Port Phillip Basin. Darragh’s molluscan assemblage zones were proposed to provide a zonation within the Southeastern Australian Province. This marine province was initially established by Crespin (1950) after she noticed that there was a distinct biogeographic discontinuity between the foraminifera of basins west of the Mount Lofty Ranges and those to the east during the Mio-Pliocene. Initially the eastern province, including the Murray, Otway, Bass, Port Phillip and Gippsland Basins was named the Bass Strait Province (Crespin, 1950) but Darragh (1985) renamed it the Southeast Australian Province.

The persistence of the E. macroptera–E. atkinsoni lineage in the Murray Basin beyond its extinction further east is evidence that the Southeast Australian Province was not biogeographically uniform in the Middle Miocene. There are other examples of persistent mollusc taxa in the Cadell Formation that are only found in older Victorian strata. The possible columbariid Hispidofusus piscatorius and the nominate subspecies of the volutid Athleta (Ternivoluta) antiscalaris are present in the Langhian aged Cadell Formation but are only found in Victorian sites that are correlated with the Burdigalian Stage (Darragh, 1969, 1971, as Batesfordian). In addition to these persistent taxa there are several endemic species in the Cadell Formation with related congeners in the Otway or Port Phillip Basins, including cypraeids (Schilder, 1935; Yates, 2008), volutids (Darragh, 1989) and a venerid (Darragh, 1965). These all suggest some form of isolation of the western Murray Basin allowing it to evolve some distinctive taxonomic differences from more easterly basins in the Province. It is interesting to note that this difference is not due to the influence of the Austral-Indo-Pacific Province to the immediate west of the Murray Basin. This province, so-named because it is an extension of the Indo-Pacific Province, shares many warm-water foraminifera and molluscs with the Indo-Pacific Province and none with the Cadell Formation that cannot be found further east. This situation was temporary and the overlying Bryant Creek Foramation of later Langhian age does contain molluscs with affinities to the Indo-Pacific Province including a species each of Nemocardium, Globularia, Sphaerocypraea and a strombid with some resemblance to Thersistrombus (A. Yates, 2009–2021, personal observations).

Conclusions

Ericusa ngayawang is a species of Ericusa, belonging to the E. macroptera–E. atkinsoni lineage. It lived in the western Murray Basin during the Langhian Stage of the Middle Miocene, when other members of the lineage had gone extinct elsewhere in the Southeastern Australian Province. The persistence of the lineage in the Western Murray Basin is matched with a few other molluscan taxa. In addition to the persistent lineages there are also some endemic mollusk species in the Cadell Formation. Together these indicate that there was some degree of biogeographic isolation of the western Murray Basin from the rest of the Southeastern Australian Province during the deposition of the Cadell Formation.

I wish to thank my family for their assistance during many collecting trips to the Cadell Formation during which the specimens described here were collected. I also wish to thank Mary-Ann Binnie (South Australian Museum) for diligently searching the collections in her care for the Ericusa specimen from Mundy’s Well, and providing data on other fossils from that locality. Chris Goudey and Angus Hawke both provided images of fine specimens of fossil Livoniini in their collections that assisted in drafting the illustrations used in this paper. Lastly I wish to thank Didier Merle, Mathias Harzhauserand an anonymous reviewer for their thoughtful and constructive suggestions which improved the quality of this paper.

Institutional abbreviations

NMV Museum Victoria, Melbourne, Australia.

NTM Museum and Art Gallery of the Northern Territory, Darwin and Alice Springs, Australia.

WAM Western Australian Museum, Perth, Australia.

Additional Information and Declarations

Competing Interests

Author Contributions

Field Study Permissions

Data Availability

New Species Registration

The author declares that they have no competing interests.

Adam M. Yates conceived and designed the experiments, performed the experiments, analyzed the data, prepared figures and/or tables, authored or reviewed drafts of the article, and approved the final draft.

The following information was supplied relating to field study approvals (i.e., approving body and any reference numbers):

The specimens were fossils collected from non-reserved public lands in South Australia. Fossils are covered by the Mining Act in South Australia which allows non-commercial fossicking without a permit provided there is no use of any mechanical equipment. Therefore there is no permit required for the collection of these specimens, neither could one be obtained.

The following information was supplied regarding data availability:

The specimens described in this article are lodged in the Palaeontological Collection of the Museum and Art Gallery of the Northern Territory.

They are physically held at Megafauna Central, 21 Todd Street, Alice Springs, 0870 Northern Territory, Australia, which is a facility run by the Museum and Art Gallery of the NT, under accession numbers: NTM P8464, P9019 and P11217.

The following information was supplied regarding the registration of a newly described species:

Publication LSID: urn:lsid:zoobank.org:pub:0CFA9C77-D089-49D5-A6FA-EA51099D57E6

Ericusa ngayawang sp. nov. Species LSID: urn:lsid:zoobank.org:act:E133FA7E-2327-4AA3-8D55-B7B9DA5CF950.

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
