# Peer review of "A new volute, Ericusa ngayawang sp. nov. (Gastropoda: Volutidae), from the Miocene of South Australia"

_PeerJ, doi:10.7717/peerj.14197_

## Round 0.1 · original submission · Minor Revisions

Illustrations should be added of the other species that are relevant for comparison: the type species of Ericusa, Voluta fulgetrum (which should be cited after line 109), Ericusa macroptera, Ericusa atkinsoni and Livonia stephensi. The last should be considered because one reviewer pointed out that Livonia stephensi might be related and perhaps would be better placed in Ericusa. If you concur, the manuscript should be revised to take this placement into account.

I do not think that all of the reviewers' comments need to be implemented, but in your rebuttal letter, you should consider each point and state your response.

One reviewer didn't find the statement about electronic publication to be necessary, but this is required by the journal, so leave it in place.

·

Basic reporting

The article is well written and correctly built. The litterature is complete regarding the context (description of a new species.
However one point needs to be clarified
1°) in the part material and method, the author gives some editor considerations on PeeerJ - These editorial considerations do not seem useful to me nor do they belong in the material and method section.
I propose to remove them.

Experimental design

The aim of the paper is to describe a new Miocene species of volutid from South Australia.
It is an original paper and the description of this species is interesting for the knowledge of the molluscan assemblages of the South Australian Miocene.
My comments are on the part Systematic Paleontology
1°) line 110 ERICUSA H. & A. Adams, 1858 => Add the type species
2°) line 124 Specific diagonsis - This part does not correspond to a diagnosis. A diagnosis should be written in telegraphic style and without too much comparisons. Thus this text is for me a good comparison. The author can change the name of the section Specific diagnosis in Comparisons and can place it below the section Description.
3°) line 124 Discussion - I find that the discussion is too long for the context of the description of a new species. It should be shortened.
4°) The raw dimensions given in the table 1 are necessary, but a diagram would be helpful to better understand potential differences.
5°) I am frustated by the figures. It would be helpful for the readers to have figures of the teleoconch and the protconch of Ericusa atkinsoni and Ericusa macroptera to better appreciate the differences with the new species.
6°) As many volutids display a colour pattern, it would be interesting to verify under UV light if the teleoconch of new species bears a residual pattern.
.

Validity of the findings

Yes, a new species of volutid is an original paper and useful for the knoledge of the family.
For the rest see above Experimental design

·

Basic reporting

There are no mistakes or flaws in this manuscript, which describes a small aspect of southern Australian paleobiogeography. Nevertheless, the paper would clearly benefit from more graphical information. Now, the reader has to rely on the taxonomic placement but could be guided by illustrations of the type species of Ericusa including a detail of the protoconch.
In the same way the discussion on related Oligocene-Miocene taxa would benefit from illustrations (e.g. macroptera atkinsoni).

Finally, the reader could follow the story much easier if the author would provide a stratigraphic chart showing all the mentioned taxa (ideally including also post Miocene occurrences) arranged in geographic columns.
Similarly, a map showing the distribution of all recent species and the occurrences of all fossil taxa would help.

Line 109. Ad type species, author, year, region.

Haviing added these minor points, the mansucript should be published as it is.



best
mathias

Experimental design

ok

Validity of the findings

ok

Additional comments

see above

Reviewer 3 ·

Basic reporting

See comments below

Experimental design

See comments below

Validity of the findings

Yates paper review
This paper describes a species which is new and should be published subject to attention to the points listed below.
It is usual to use a telegraphic style in the actual description of the taxon.e.g. Shell small for Ericusa (Table 1), estimated adult length approximately 70mm, holotype lying within known size range of Ericusa subtilis, exceeding only…… and so on.
Under discussion the author should look at and compare his taxon with Livonia stephensi (Johnston, 1880), which has some similarity in morphology to it. Note that Darragh (1989, p. 252) in referring to L. stephensi stated that ‘the protoconch is more like that found in species of Ericusa’ and only placed it Livonia because of resemblance to some other species of Livonia. If the author’s specimens of his new species had worn protoconchs they would look very like L. stephensi in overall morphology, suggesting that maybe stephinsi would be better placed in Ericusa. This would have implications for the author’s discussion on lineage of his new species.
When palaeontological specimens are coated lightly with ammonium chloride or magnesium oxide before photography the resulting images are fare superior to images of specimens without such coating. This is the norm for palaeontological papers. The author should consider using this technique.

---

## Round 0.2 · Minor Revisions

Thank you for your careful consideration of the reviewers comments and the modifications to the manuscript that you made in response to them.

The species is appropriately described and interesting. Naming it after a tribe/region is highly commendable and appropriate but please provide some more context (e.g., references for the etymology of this land/tribe).

At your discretion, you might also consider adding some more words on what happened to this region/tribe and the personal reason behind naming it that way.

---

## Round 0.3 · accepted · Accept

Thank you for adding the statement of etymology.